# Seroprevalence of dengue virus antibodies among multiple species of non-human primates in Senegal suggests that sylvatic dengue virus is maintained in non-primate reservoirs in this region

Stephanie C. Cinkovich[1,2]☯, Benjamin M. Althouse [3,4]☯*, Matt D.T. Hitchings[2,5], Prudny Bonnaire-Fils[5], Ousmane M. Diop[6], Ousmane Faye[6], El Hadji Abdourahmane Faye[6], Diawo Diallo[6], Bakary Djilocalisse Sadio[7], Abdourahmane Sow[6], Oumar Faye[6], Mawlouth Diallo[6], Brenda Benefit[7], Douglas M. Watts[8,9], Amadou A. Sall[6], Scott C. Weaver[9,10,11], Kathryn A. Hanley[4], Derek A.T. Cummings[1,2,12,13]

**1** Department of Biology, University of Florida, Gainesville, Florida, United States of America, **2** Emerging Pathogens Institute, University of Florida, Gainesville, Florida, United States of America, **3** Information School, University of Washington, Seattle, Washington, United States of America, **4** Department of Biology, New Mexico State University, Las Cruces, New Mexico, United States of America, **5** Department of Biostatistics, College of Public Health & Health Professions, University of Florida, Gainesville, Florida, United States of America, **6** Institut Pasteur de Dakar, Dakar, Senegal, **7** Department of Anthropology, New Mexico State University, Las Cruces, New Mexico, United States of America, **8** Office of Research and Sponsored Projects, University of Texas at El Paso, El Paso, Texas, United States of America, **9** Center for Biodefense and Emerging Infectious Diseases and Department of Pathology, University of Texas Medical Branch, Galveston, Texas, United States of America, **10** Department of Microbiology and Immunology, University of Texas Medical Branch, Galveston, Texas, United States of America, **11** Institute for Human Infections and Immunity, University of Texas Medical Branch, Galveston, Texas, United States of America, **12** Department of Epidemiology, Bloomberg School of Public Health, Johns Hopkins University, Baltimore, Maryland, United States of America, **13** Department of Biomedical Engineering, Whiting School of Engineering, Johns Hopkins University, Baltimore, Maryland, United States of America

☯ These authors contributed equally to this work.
* bma85@uw.edu

## Abstract

Dengue virus (DENV) circulates in two distinct transmission cycles: one, termed the sylvatic cycle, is enzootic to canopy-living hosts, including non-human primates and primatophilic mosquitoes, and the other, initiated by spillover from the sylvatic cycle, is endemic to humans and anthropophilic mosquitoes. Transmission dynamics of sylvatic DENV in non-human hosts has not been well characterized, and the identity of reservoir and amplification hosts is still to be determined. We investigated the role of the three common species of monkeys in the Kédougou region of Senegal in the sylvatic transmission cycle of DENV. Longitudinal surveillance of primatophilic mosquitoes in this region dating back to the 1970s revealed that sylvatic DENV-2, the only one of the four DENV serotypes found to circulate in a sylvatic transmission in West Africa, is amplified cyclically at intervals of approximately eight years based on the isolation of the virus from mosquitoes. Subsequent to the detection of DENV-2 in

**Data availability statement:** All relevant data are available at doi: https://doi.org/10.5281/zenodo.15786080.

**Funding:** This work was supported by grants 1R15AI113628-01 (KAH), 1R01AI145918-02 (KAH, BMA), RO1AI069145 (BMA, SCW, KAH), R24AI120942 (SCW), 1U01AI115577-01 (SCW, KAH) from the U.S. National Institutes of Health. The funders had no role in study design, data collection and analysis, decision to publish, or preparation of the manuscript.

**Competing interests:** The authors have declared that no competing interests exist.

primatophilic mosquitoes in Kédougou in 2008, 737 monkeys, including 3 species: *Chlorocebus sabaeus* (n = 219), *Erythrocebus patas* (n = 78), and *Papio papio* (n = 440) were captured from 2010 to 2012 for the current study. Their age was determined using dentition and other morphological measurements. Evidence of DENV-2 infection was detected via neutralizing antibody in sera, and the annual hazard of DENV-2 infection was estimated per species using catalytic models. These analyses revealed annual hazard ranging from 0.09 to 0.42 across the three species, consistent with high levels of transmission in these populations. Furthermore, seroprevalence was moderate in individuals under one year of age, despite the lack of detection of DENV-2 in primatophilic mosquitoes for up to three years prior, suggesting that non-primate hosts contributed to the maintenance of sylvatic DENV in this region.

## Author summary

Dengue is often viewed as a disease that circulates mainly among people in cities, but in West Africa the virus also persists in forest environments, where it infects wild mosquitoes and animals and can occasionally spill over into humans. We set out to better understand the role of non-human primates in this forest, or sylvatic, transmission cycle. To test this idea, we collected blood samples from over 700 monkeys of three different species in southeastern Senegal between 2010 and 2012 and measured antibodies that indicate past dengue infection. We found that a large proportion of monkeys had been infected, including many young animals. Notably, young monkeys showed evidence of infection even during years when dengue virus was not detected in mosquitoes, suggesting that transmission continues quietly in the environment. We also observed clear differences in infection rates between monkey species, pointing to ecological or behavioral factors that influence exposure. Together, our findings suggest that monkeys alone are unlikely to fully maintain dengue virus transmission and that other animal hosts may also contribute. Understanding how dengue virus persists outside human populations is important, because environmental change and expanding human activity near forests may increase the risk of spillover and future outbreaks.

## Introduction

The four serotypes of dengue virus (DENV-1–4) emerged from sylvatic cycles in the forest canopy of tropical Asia into human populations [1–4]. Canopy-living non-human primates (NHPs) have been shown to be seropositive for all four DENV serotypes in Asia, DENV has been isolated from canopy-living mosquitoes as well as sentinel macaques placed into the forest canopy in Asia [2,5,6], and DENV has been isolated from NHPs in West Africa [7,8]. Together, this evidence has prompted the widespread, but thinly supported, belief that NHPs are the reservoir of sylvatic DENV [9]. Furthermore, limited evidence of a reservoir role is supported by recent

experimental studies that revealed extremely low replication of sylvatic DENV-2 in both cynomolgus macaques (*Macaca fascicularis*) [10] and African green monkeys (*Chlorocebus sabaeus*) [11] and low transmission of sylvatic DENV-2 from cynomolgus macaques to mosquitoes [10], calling this belief into question.

Alone among the four DENV serotypes, DENV-2 has established a sylvatic cycle in Africa, although apparently limited to West Africa. Sylvatic DENV-2 was first isolated in 1970 in Senegal, where longitudinal surveillance of primatophilic mosquitoes revealed periodic, presumably epizootic amplifications with silent intervals of 5–8 years [12–17]. Sylvatic DENV amplifications tend to occur at the end of the rainy season and may be sensitive to temperature and circulation of other flaviviruses [12]. The hypotheses for this periodic behavior have included amplifying host population turnover and herd immunity in NHP populations [1,13], and mathematical modeling showed that the dynamics in this system were consistent with long-cycle periodicity driven by primate population turnover and the build-up of herd immunity together with periodic stochastic reintroduction events perhaps driven by a faster reproducing, but non-surveilled, reservoir species [14]. However, serosurveys for chikungunya virus (CHIKV), which is transmitted in sylvatic habitats by the same mosquito species as DENV, conducted in NHPs in southern Senegal found high seroprevalence in infants during silent intervals, implying that there is sustained transmission even in years with no isolation of virus from mosquitoes [15].

Sylvatic DENV-2 in West Africa has spilled over into humans on multiple occasions [1,16–20], including infection of one individual who carried the virus to Europe [18]. Additionally, two highly divergent sylvatic lineages were detected in travelers arriving in Australia from Borneo in 2014 (DENV-1) and 2015 (DENV-2) [21–23]. These cases demonstrated the ability of sylvatic DENV to cause life-threatening symptoms in humans and to be transported across continents [18,21–25]. No significant adaptive barrier to the emergence of sylvatic DENV into human populations exists [26–29]; thus, proximity to forest habitat is likely to be the key risk factor for sylvatic spillover and subsequent human-amplified outbreaks [1,30,31]. Such spillovers may accelerate in West Africa, which is experiencing rising rates of natural habitat encroachment by humans and fragmentation [19,24,32]. Regional climate projections indicate a northward and altitudinal expansion of *Aedes aegypti*-suitable habitat across the West and Central Africa by mid-century, further tightening the sylvatic–urban interface [33]. Furthermore, the evidence of human endemic DENV-1, -2, and -3 across Senegal increases the potential risk for forming hypering-endemic DENV communitites to increase the risk of more severe outcomes from the possibility of antibody-dependent enhancement occurring among heterotypic infections [34,35].

A better understanding of the ecology of sylvatic DENV in West Africa may inform risk factors for human populations and control strategies. Moreover, characterizing the transmission dynamics of sylvatic DENV, including identifying the roles played by different species [15], will be crucial for quantifying the risk of re-emergence [36,37]. Recent evidence of a cross-protective effect of antibodies, resulting from sylvatic DENV infection, against yellow fever virus (YFV) also demonstrates the potential impact of sylvatic DENV circulation on other arthropod-borne viruses [38]. Thus, similar to our previous study on CHIKV, we hypothesized that monkeys were reservoir hosts for sylvatic DENV in Kédougou, southeastern Senegal, and therefore that the periodic amplification of sylvatic DENV detected in primatophilic mosquitoes was regulated by depletion of susceptible NHP hosts during epizootics (epidemics in the reservoir hosts), local extinction of the virus, recruitment of susceptible hosts via births, and reintroduction of the virus from NHP populations at distant sites. We tested this hypothesis using data from a three-year age-stratified serosurvey for sylvatic DENV-2 of the three main NHP species present in southeastern Senegal to characterize the rate of acquisition DENV and determine whether the serostatus of these NHPs was consistent with sustained transmission in NHPs. Age-stratified seroprevalence was used to estimate the force of infection and basic reproductive number, $R_0$, of sylvatic DENV-2 in these populations.

## Methods

### Ethics statement

All animal research was approved by the Institutional Animal Care and Use Committee (IACUC) of University of Texas Medical Branch, Galveston, protocol number: 0809063 (principal investigator: S.C.W.), and the entire protocol was

approved on 27 November 2008 by the Consultative Committee for Ethics and Animal Experimentation of the Interstate School for Veterinary Sciences and Medicine, Dakar, Senegal (principal investigator: A.A.S.). No other specific permits were necessary. This approval is necessary and sufficient to conduct wildlife research in Senegal.

Methods employed in this study are described in Althouse et al. 2018 [15]. Brief summaries are presented here.

## Study site

NHPs were trapped at various sites around the Department of Kédougou in the southeast region of Senegal. The Kédougou region comprises diverse land cover types spanning open savanna to gallery forests that exist along valleys and rivers. The tropical savanna climate of Kédougou has one rainy season from May to November with an average of 51 inches of rainfall and has year-round temperatures of 25–33°C [39]. Three monkey species occur in Kédougou, all of which exist as Least Concern or Near Threatened IUCN (International Union for Conservation of Nature and Natural Resources) classification: African green monkeys (*Chlorocebus sabaeus*), patas monkeys (*Erythrocebus patas*), and Guinea baboons (*Papio papio*) [40]. A population of chimpanzees (*Pan troglodytes*) is also present, but in lower numbers [40]. Population sizes for resident Senegal bushbabies (*Galago senegalensis*) are unknown and, because of their cryptic nocturnal behavior, they were not collected for this study. While human populations in Kédougou have been historically small and dispersed (four people/km$^2$), a recent increase in gold-mining activities, sparked by foreign industrial mining companies, has increased the population density and mobility characteristics of this region [41].

## Protection of subjects

Trapping and sampling collection methods were approved by the Institutional Animal Care and Use Committee (IACUC) of University of Texas Medical Branch, Galveston, protocol number: 0809063 (principal investigator: SCW), and the protocol was approved on November 27, 2008 by the Consultative Committee for Ethics and Animal Experimentation of the Interstate School for Veterinary Sciences and Medicine, Dakar, Senegal (principal investigator: OD). Animals were trapped in large, open air cages using food bait with access to water and were then sedated using ketamine and retained only long enough to take the necessary anthropomorphic measurements, collect a blood sample, and to ensure they were released as an intact and healthy troop upon recovery from anesthesia. Capture sessions were organized by trapping troops at specific sites during defined time windows, and individuals were identified in the field by a combination of species, sex, estimated age class, body size, and unique physical characteristics (e.g., scars, pelage markings). Animals were not individually tagged and were released at the site of capture after sampling. While we cannot absolutely exclude the possibility of recapturing a small number of individuals across years, the geographic separation of trapping sites, the temporal spacing of campaigns, and the large populations and vast home ranges of these three species in this region together served to minimize repeat sampling. Moreover rare resampling is unlikely to materially affect our reported age-stratified seroprevalence patterns we report.

## Serology

Monkeys were bled from the inguinal vein and sera were separated and frozen. Sera were tested for DENV-2 antibody by a plaque reduction neutralization tests (PRNT) performed in 12-well plates with confluent Vero cell cultures using 800 focus-forming units of DENV-2 strain 16681, a 1964 human isolate from Thailand (GenBank Accession No. U87411.1). This strain was used because it plaques efficiently and cross-reacts extensively with all other DENV-2 isolates including sylvatic strains from Senegal. Serial dilutions of sera were mixed with an equal volume of virus and incubated for one hour at 37°C. A 0.25 mL volume of each serum–virus mixture was then added to wells containing Vero cells and incubated for one hour at 37°C, 5% $CO_2$. After adsorption, 1.0 mL of 4% methylcellulose in OPTIMEM-I overlay (GIBCO BRL, Gaithersburg, MD) was added to each well and plates were incubated for four days at 37°C, 5% $CO_2$. Plates were fixed with 1:1 methanol:acetone, and foci were stained with peroxidase-conjugated anti-DENV-2 antibody

and counted to determine the maximum serum dilution that neutralized 50% or 80% of foci. Neutralizing antibody titers were defined as the reciprocal of the highest serum dilution resulting in a 50% or 80% reduction in foci relative to virus-only controls ($PRNT_{50}$ and $PRNT_{80}$, respectively), and sera with $PRNT_{50}$ titers ≥ 1:20 or $PRNT_{80}$ titers ≥ 1:20 were considered seropositive for prior DENV-2 infection. As part of the broader arbovirus study, yellow fever virus (YFV) $PRNT_{90}$ assays were also performed on a subset of sera from this cohort; we summarize these results descriptively but did not incorporate them into the DENV-2 force-of-infection models because the number of animals with informative YFV titers and potential cross-reactive patterns was limited. The samples described in this study were used in an analysis of sylvatic CHIKV virus transmission [15].

## Determination of NHP Age

The collected monkey species (*C. sabaeus, E. patas,* and *P. papio*) were categorized into age classes based on an algorithm that utilized measurements of tooth eruption and degree of molar wear. The age class was estimated by combining tooth eruption sequences and gingival eruption information collected from dental casts and photographs, with published ages of dental eruption based on individuals of known age and gender from captive populations [42–45]. For those with inadequate dental casts, anthropomorphic measurements, including weight, coloration, reproductive organ development, and limb lengths, were used to estimate their age. For more information on aging methods and age classes for NHPs used in this study, view the supplemental information associated with Althouse et al. 2018 [15].

## Force of infection

As an indicator of transmission intensity, the force of infection (FOI) provided estimates of the hazard of susceptible individuals becoming infected over time. The degree to which seropositivity increases with age informs the rate at which susceptible individuals acquire infection at certain ages. To determine the annual forces of infection ($\lambda$), catalytic models of infection were fit to age-stratified data by primate capture per year. Model equations were adapted from Grenfell *et al.* [15,46]. The proportion of the population susceptible to DENV-2 infection of age $a$ at time t is given by $x(a,t) = e^{-\int_0^a \lambda(t-\tau)d\tau}$. The proportion of individuals of age $a$ infected with DENV-2 at time t is $z(a,t) = 1 - x(a,t)$. We assumed that $\lambda$ was constant across age and time, and we estimated $\lambda$ within species and year of collection to detect changes across calendar time. We discretized NHP age by year into age classes $a_k$, $k \in [1, m]$, and estimated $\lambda$ by maximizing the likelihood. The binomial log-likelihood of $\lambda$ is $L(\lambda) = \sum_{k=1}^{m} [n_{xk}\log[x(a_k,t)] + n_{yk}\log[z(a_k,t)]]$, where $n_{xk}$ and $n_{yk}$ were the number of seronegative and seropositive for DENV-2 infection in age class $k$, respectively. We estimated $\lambda$ for each species and year separately, using $PRNT_{80}$ seropositivity to define whether an individual had been infected. $PRNT_{50}$ seropositivity was used as a secondary definition of infection. To estimate uncertainty in estimates of $\lambda$, bootstrap confidence intervals were calculated by sampling NHPs with replacement and recalculating $\lambda$.

## Mixed effects logistic regression

Associations between NHP attributes and DENV-2 seropositivity were explored using a mixed-effects logistic regression. Outcomes of $PRNT_{80}$ and $PRNT_{50}$ seropositivity were investigated in this framework. The covariates of interest were NHP age, month of collection, year of collection, and species, with NHP troop (same site and collection date) as a random effect to account for correlation at the troop level. All analyses were conducted in R version 4.4.3.

## Estimating the basic reproductive number of sylvatic DENV-2

The basic reproductive number ($R_0$) represents the number of secondary infections caused by one infectious individual in a wholly susceptible population. $R_0$ was calculated by using the age-specific hazard, $\lambda$, to estimate the fraction of the population that remained susceptible, together with assumptions about the age structure of the population and an assumption that the age structure of seropositivity was at equilibrium. When $f(a)$ is the fraction of the population of age $a$ and

$w(a,t)$ is the fraction of the population of age $a$ exposed to DENV-2 at time $t$, then $R_0 = \frac{1}{1-\int_0^\infty f(a)w(a,t)da}$, where $w(a,t)$ was estimated from the age-specific $\lambda$ as $w(a,t) = 1 - e^{-\int_0^a \lambda(t-\tau)d\tau}$. The true age structure of the NHP populations surveyed was unknown. We assumed that the age-distribution of the population follows an exponential distribution with a mortality rate = 1/ mean observed age.

## Results

### Age, gender, and species breakdown

Throughout the study (2010–2012) a total of 737 NHPs were collected over 15 sites. This included 219 *C. sabaeus,* 78 *E. patas* (patas monkeys)*,* and 440 *P. papio* (Guinea baboons), where the minimum number of NHPs collected by species in a given year was four (2011, *E. patas*) and the maximum number collected was 200 (2011, *P. papio*). More males were collected for both *C. sabaeus* and *P. papio,* 147 versus 70 and 260 versus 180, respectively. The mean ages for each species were 4.0 years for *C. sabaeus*, 6.7 years in *P. papio*, and 3.5 years in *E. patas* [15], with some variation in age distribution across years for *E. patas* in particular.

### Serology

A total of 81 NHPs were not tested due to inadequate collection of (i) a blood sample, (ii) anthropomorphic data, (iii) dental casts and photographs, or (iv) samples being lost during shipment. Of the remaining 656 NHPs (198 *C. sabaeus*, 399 *P. papio*, and 59 *E. patas*), seropositivity rates for DENV across all three species were high: 344 (52%) and 477 (73%) were seropositive for DENV by $PRNT_{80}$ and $PRNT_{50}$, respectively. Maximum $PRNT_{80}$ titers of 1:640 were observed, and the distributions of $PRNT_{80}$ titers in each year were unimodal (Fig 1).

 YFV $PRNT_{90}$ assays were available for 298 NHPs in this cohort (94 *C. sabaeus*, 7 *E. patas*, and 197 *P. papio*). Among these, 176 animals had any evidence of YFV $PRNT_{90}$ reactivity, and age-seroprevalence patterns did not demonstrate a clear increase with age (S1 Fig). There was a significant association between DENV-2 and YFV seropositivity ($\chi^2$ test p < 0.001), which may indicate cross-reactivity or, alternatively, heterogeneity in exposure to arboviruses. A total of 67 subjects had a recorded quantitative YFV $PRNT_{90}$ titer. However, titers among these 67 animals were generally low (76.1% had YFV titers less than or equal to 80, 16.4% between >80 and ≤160, and 1.5% greater than 320) and did not define a clear YFV-exposed group. Given the paucity of $PRNT_{90}$-positive animals under our prespecified criteria and the relatively

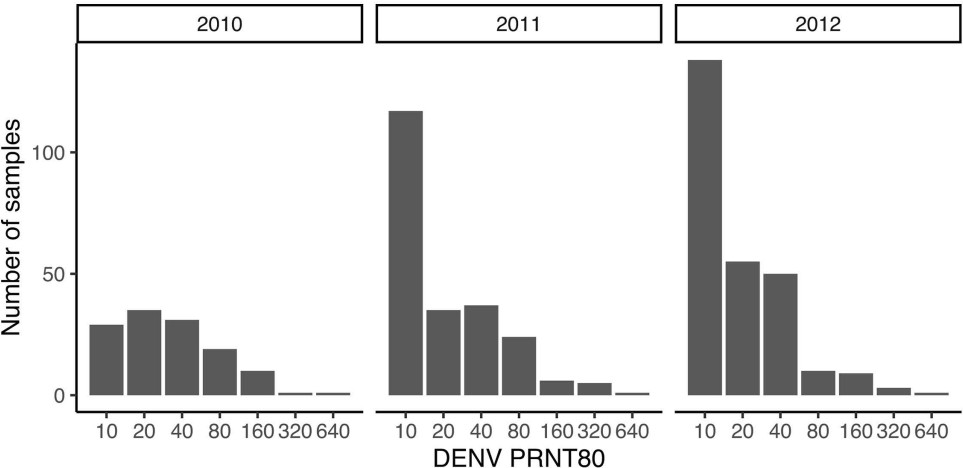

**Fig 1. Distribution of DENV $PRNT_{80}$ titers by year in monkey sera collected in the Kédougou region, southeastern Senegal, 2010-2012.**

small number with detectable titers, we did not attempt to jointly model YFV and DENV-2 forces of infection and instead presented the YFV results descriptively as contextual information.

## Quantifying the transmission of Sylvatic DENV-2

The force of infection (FOI) on all NHP species during each year of this study (except 2011 *E. patas*) demonstrated high rates of infection by sylvatic DENV-2 in young age classes, which was not an artifact of maternal antibodies, as a continued positive trend of seropositivity was seen with age and ages beyond the first year of life when maternal antibody was expected to be present (Fig 2). *C. sabaeus* had the highest estimated FOI (0.40, 95% CI, 0.32,0.53 averaged across the three study years), with low annual variation. Estimates for *E. patas* were lower at 0.17 (95% CI, 0.10,0.30) averaged across the study period. *P. papio* appeared to have the lowest FOI over the three-year study, as the proportion of seropositive *P. papio* did not plateau within their lifespan. When collapsing all years of surveillance down to one overall FOI estimate, the *C. sabaeus* had at least double the FOI compared to the two other species (Fig 2). When using $PRNT_{50}$ instead of $PRNT_{80}$ seropositivity to define prior infection, estimated FOIs were higher (S2 Fig), and there was greater annual variation in the FOI.

Species differences in FOI were supported by a mixed-effects logistic regression for DENV-2 $PRNT_{80}$ seropositivity, with the troop (same collection year and site) as a random effect (Table 1). In this analysis, *E. patas* and *P. papio* had significantly lower DENV-2 seropositivity relative to *C. sabaeus*, adjusting for collection year and age. The intraclass correlation of the random effect model indicated that 9.3% of the total observed variance in DENV-2 seropositivity was due to variance among NHP troops. These associations were qualitatively similar when using DENV-2 $PRNT_{50}$ seropositivity as the outcome (S1 Table).

$R_0$ estimates were consistently greater than one for all three years and species for which there were sufficient data (Table 2). Estimates of $R_0$ varied from 1.58 (95% CI, 1.29,2.24) in *E. patas* in 2012 to 2.98 (95% CI, 2.39,4.04) in *P. papio* in 2010. $R_0$ estimates were higher when using $PRNT_{50}$ seropositivity to define prior infection (S2 Table), and remained significantly greater than one for all species and years.

## Discussion

Annual collection and testing of pools of primatophilic mosquitoes have revealed 5- to 8-year amplification cycles of sylvatic DENV-2 transmission in southeastern Senegal [14]. Detection of DENV-2 in sylvatic mosquito vectors in 2008 [12] together with previous modeling predicting vanishingly low levels of transmission among primates in mosquito detection-silent intervals [14] suggested that there should have been a build-up of susceptible individuals from 2010 to 2012 in NHP populations. However, the high seroprevalence among these three NHP species (52% of 656 NHP sampled), even in infants, and corresponding high FOI, contradicts this hypothesis. Instead, consistent with findings for CHIKV seropositivity in these species [15], our study suggested that infants were at risk of infection even in silent years, and that DENV-2 incidence of infection may be consistently high in infant NHPs, reflecting a spillover risk to humans.

We found high variability in age-seroprevalence curves among species, and some variation among troops, with significantly higher seropositivity in *C. sabaeus* as compared to *E. patas* and *P. papio*. These differences may reflect differences in troop size, with larger troops able to sustain more transmission, or differences in the propensity of species to be bitten by mosquitoes that transmit DENV and to be in proximity of other hosts. Consistent with published primatology data, our captures reflected typical troop sizes of approximately 3–19 individuals for *C. sabaeus* (median 8) [47], 1–16 individuals for *E. patas* (median 7) [48], and 5–64 individuals for *P. papio* (median 31) [49] across distinct troops. These differences in group size, together with differences in daily ranging distances (*P. papio* > *E.patas* > *C. sabaeus*) and, probably most importantly, arboreality (*C. sabaeus* > *P. papil* > *E.patas*) likely influence exposure to primatophilic mosquito vectors and may contribute to the higher sylvatic DENV-2 FOI we observed in *C. sabaeus* compared with *E. patas* and *P. papio*. Several studies have similarly identified variability: for example, a serosurvey of monkeys in Kenya found DENV IgG

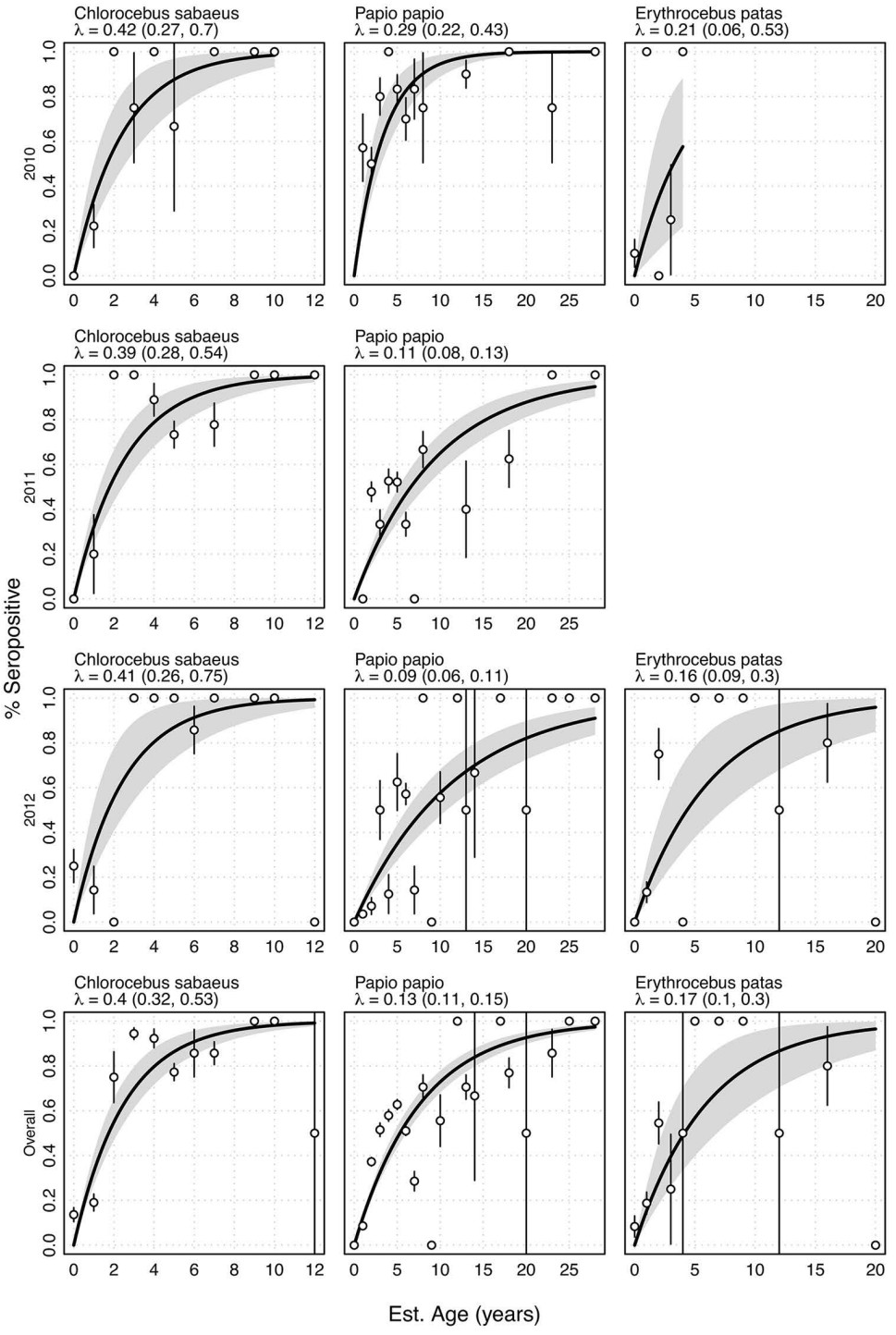

**Fig 2. Forces of infection (FOI) by species and year where panels show the forces of infection, $\lambda(a)$, for *C. sabaeus, P. papio*, and *E. patas*, respectively.** There were too few *E. patas* collected in 2011 to estimate FOI. The last panel shows an overall FOI for each species, using all years of data for the single estimate. All monkeys under one year of age were grouped into the '0' age category. Points represent the proportion of seropositive animals per age and year, with associated confidence interval as vertical lines. Seroprevalence estimated from model using best fitting forces of infection (black line), with grey shaded bands representing bootstrap 95% confidence intervals.

**Table 1. Mixed effects logistic regression for DENV seropositivity in monkey samples collected in 2010-2012, in the Kédougou region, southeastern Senegal.** Estimates from a mixed effects model with DENV PRNT$_{80}$ seropositivity as the outcome and monkey age, species, and year of collection as mixed effects with troop (same collection date and site) as the random effect.

| Covariate | Odds Ratio (95% CI) |
|---|---|
| Age | 1.20 (1.14,1.27) |
| Species (vs. *Chlorocebus sabaeus*) | |
| *Papio papio* | 0.27 (0.13,0.58) |
| *Erythrocebus patas* | 0.25 (0.08,0.72) |
| Collection Year (vs. 2010) | |
| 2011 | 0.48 (0.18,1.26) |
| 2012 | 0.46 (0.18,1.19) |
| Month captured (vs. January) | |
| February | 1.69 (0.36,7.93) |
| March | 1.07 (0.33,3.50) |
| April | 1.68 (0.44,6.46) |
| May | 1.91 (0.45,8.12) |
| December | 0.55 (0.08,3.94) |
| Random Effect | Estimate |
| Random Effect Standard Deviation - Troop | 0.58 (0.00,0.81) |
| Intraclass Correlation Coefficient | 0.093 |

**Table 2. Estimates of $R_0$ for each NHP species by year.** The age distribution is assumed to be exponentially structured with rates equal to the mean lifespans reported in this study.

| Year | *C. sabaeus* $R_0$ (95% CI) | *P. papio* $R_0$ (95% CI) | *E. patas* $R_0$ (95% CI) |
|---|---|---|---|
| 2010 | 2.74 (2.01,4.12) | 2.98 (2.39,4.04) | 1.77 (1.14,3.38) |
| 2011 | 2.55 (2.09,3.41) | 1.71 (1.53,1.94) | NA |
| 2012 | 2.66 (1.96,4.50) | 1.58 (1.42,1.80) | 1.58 (1.29,2.24) |

seropositivity of 41% and 15% in two species of *Papio* baboons compared to 0% among monkeys in the *Cercopithecus* genus [50]. On the other hand, a previous study in Kédougou found similar levels of DENV-2 seropositivity in *C. sabaeus* and *E. patas* [51]. Thus, across the literature, there is little consistency in which NHP species has higher DENV seropositivity. This lack of a common trend may be due to the small number of studies comparing multiple species in the same location, and small sample sizes within studies leading to increased variability, or it may reflect real stochasticity in sylvatic DENV transmission.

Overall estimates of seropositivity obtained in these populations (52% by PRNT$_{80}$) were within the rather wide range observed in other NHP populations, including from West Africa; a serosurvey of NHPs in Kédougou between 2002 and 2006 found DENV-2 IgG seroprevalence of 16% and 21.4% among *C. sabaeus* and *E. patas,* respectively, with high inter-annual variability [51]. A 1999 serosurvey of *C. sabaeus* in Senegal found 58% seroprevalence for DENV-2 IgG [52], and two serosurveys of galagos in Nigeria found 48.9% and 25% seropositivity [53], while a serosurvey of mandrills across several countries in Central Africa found seropositivity of 8% [54]. These differences may reflect higher circulation of sylvatic DENV-2 in Kédougou compared to the other locations sampled, consistent with human IgG seroprevalence to anti-DENV envelope protein, and neutralizing antibodies [55]. However, previous studies generally used ELISA to detect

DENV IgG antibody, which has lower specificity than the PRNT [56]. The age distribution of captured NHPs could also impact seroprevalence across studies, as could the ecological niche of the species [54].

Maternally transferred DENV antibodies could explain high seropositivity in infants, but these antibodies likely decay during the first year of life [57–60], and the pattern of seropositivity in NHPs aged ≥1 year is consistent with a high FOI. Another possible explanation for these findings is transmission between distinct populations of NHPs, with periodic stochastic introduction events leading to outbreaks, although the unknown population sizes and connectivity of NHP troops makes this hypothesis difficult to assess [14].

Our data are most consistent with frequent exposure of non-human primates to sylvatic DENV-2 in Kédougou, with infants in all three species acquiring infection rapidly even in years when virus is not detected in primatophilic mosquitoes. These findings, together with prior modeling work, support the view that NHPs alone are unlikely to sustain continuous transmission but do not in themselves demonstrate cryptic sylvatic circulation of other DENV serotypes. Extensive annual arbovirus surveillance by Institut Pasteur de Dakar in this region has to date only detected sylvatic DENV-2, and we therefore interpret our serologic and modeling results specifically in the context of sylvatic DENV-2.

An additional limitation is that our serologic assays cannot distinguish between antibodies generated by strictly sylvatic DENV-2 transmission and antibodies generated following occasional spillback of human-endemic DENV-2 into NHPs. Our trapping sites were located in forested areas where the known sylvatic DENV-2 cycle was active, and the timing of our age-specific seroprevalence patterns followed periods of documented sylvatic amplification in mosquito surveillance. Moreover, sustained human dengue virus transmission was not documented in the immediate vicinity of these sites during the study period. Finally, sylvatic DENV-2 has been isolated from a patas monkey in this region [20]. Thus we believe that spillback of DENV-2 from humans, if it occurs at all, is likely rare and therefore would have minimal impact on our findings.

Finally, we did not perform Zika virus (ZIKV) PRNTs in this cohort because validated ZIKV neutralization assays were not yet available at the time of sample processing and resources were prioritized toward arboviruses known to be actively circulating in the region (including DENV-2, YFV, and CHIKV). YFV PRNTs were performed only on a subset of the data, limiting our ability to parameterize a model of cross-reactivity. As a result, we cannot directly assess potential ZIKV exposure or quantify ZIKV's and YFV's contribution to flavivirus cross-reactivity in our DENV-2 PRNTs. We therefore interpret our DENV-2 serologic results with caution, noting that unmeasured ZIKV or other flavivirus infections could contribute to residual cross-neutralization in a small subset of sera and overestimation of sylvatic DENV-2 FOI. Future studies using multi-flavivirus panels including ZIKV will increase our understanding not only of sylvatic DENV transmission but also of important arboviruses with spillover risk.

In West Africa, DENV outbreaks have been observed in Senegal, Burkina Faso, and Côte d'Ivoire, while detection of DENV in vectors in Nigeria and Guinea has also occurred [61,62]. Risk may not be homogenous throughout the region, but the right ecological conditions and migration can set the stage for outbreaks [63]. In Senegal, DENV is hyperendemic [64], and an urban epidemic of DENV-3 in 2009 reflects an epidemiological shift in Senegal away from occasional spillover outbreaks of sylvatic DENV-2 infection in rural areas to more sustained human-to-human outbreaks with transmission by *Ae. aegypti* [65,66]. More recent years have seen multifocal outbreaks with DENV-1–3 and an epidemic stemming from the Grand Magal Pilgrimage in 2018 [34,65,67]. Nonetheless, sylvatic DENV continues to pose a threat in Senegal and neighboring countries, where spillover into the human transmission cycle can trigger outbreaks [19,31]. A description of a 2020 outbreak of sylvatic DENV-2 in Kédougou, Senegal [17] demonstrated increased risk of exposure among young men 15–45 years old working outside, suggesting that exposure to mosquitoes is a major risk factor for acquisition of sylvatic DENV.

The characterization of reservoir hosts is just one component of understanding sylvatic DENV risk in humans. The Kédougou region boasts a diverse mosquito population, where *Ae. furcifer*, *Ae. vitattus*, *Ae. taylori*, and *Ae. luteocephalus* have been implicated as important sylvatic vectors and *Ae. aegypti* as the primary urban vector [52,61,68,69]. *Aedes furcifer* and *Ae. luteocephalus* are highly susceptible to both sylvatic and urban strains of DENV-2 in experimental

transmission studies, while *Ae. vitattus* and peridomestic *Ae. aegypti* are refractory to both strains [70]. Additional laboratory experiments demonstrated that the urban dwelling *Ae. aegypti* were highly susceptible to endemic DENV-2 strains, but were significantly less susceptible to sylvatic DENV-2 strains [71]. Further, despite high disseminated infection rates for urban and sylvatic *Aedes* species exposed to DENV-1, DENV-3, and DENV-4 in Senegal, low potential transmission rates reflected their low transmission capacity for these subtypes [68]. Both domesticated *Ae. aegypti aegypti* and sylvatic *Ae. aegypti formosus* morphological forms can be found throughout Senegal, but inter-family heterogeneity makes the distinction into *Ae. aegypti* subspecies invalid in the country [72]. Specific phenotypic patterns and behavioral differences that are common to the *Ae. aegypti* group in other regions are not as well-defined in West African populations, making identification of *Ae. aegypti* subspecies problematic in this area [73].

Our new analysis of age-stratified seroprevalence to sylvatic DENV-2 and the study of CHIKV [15] in the same samples from these three primate species paint a remarkably similar epidemiological picture but at the same time show key differences. Both viruses share a striking baseline: our current DENV-2 results and the CHIKV results each revealed that a substantial proportion of infant NHPs already carry neutralizing antibodies, implying steep forces of infection ($R_0 > 1$) and, taken with mechanistic modeling of transmission in these hosts, suggest that it is unlikely that these three species of monkeys sustain continuous transmission in this area. Yet, the similarities end there. For DENV-2, transmission intensity is strongly species-stratified—highest and earliest in African green monkeys, moderate in patas monkeys, lowest in baboons—whereas CHIKV infected all three hosts at comparable rates. Studies of vector-host interactions are consistent with this observation, with the DENV vector *Ae. taylori* showing a preference for African green monkeys [74], and CHIKV being transmitted by a wider variety of vectors than DENV [75], underscoring a virus-specific blend of vector preference and host competence.

Climatic signatures also diverge: an analysis of a 50-year entomological time series showed that higher rainfall and temperature depress DENV isolations from mosquitoes, while CHIKV isolations exhibit no consistent weather response [12]. This environmental asymmetry mirrors their multi-annual rhythms—DENV-2 spillover epizootics recur every ≈5–8 years at the close of the rainy season, whereas CHIKV spillover flares on shorter ≈4-year cycles spanning a broader seasonal window—implying that host-demographic replenishment and vector seasonality constrain DENV more tightly. Finally, long-term surveillance detected modest interference between viruses—years with abundant yellow-fever activity suppress both DENV circulation and, to a lesser extent, CHIKV—suggesting that inter-virus competition, although present, shapes each pathogen's landscape differently [12]. Taken together, the paired datasets highlight a common backdrop of intense, cryptic arbovirus transmission while stressing that the host, climate, and competitive ecology governing spill-over risk are virus-specific and cannot be generalized from one sylvatic arbovirus to another.

Demographic and land-use shifts amplify the urgency to understand sylvatic arbovirus spillover risk. Rapid population influx driven by gold-mining and agricultural expansion is reshaping the Kédougou landscape, increasing human–vector contact along forest edges just as opportunistic *Aedes* vectors exploit anthropogenic oviposition sites. An impact of gold mining on risk of CHIKV in the region has already been documented [76]. The same ecological churn that facilitates monkey-to-human transmission also creates corridors for human-imported serotypes to seed new sylvatic foci via spillback, potentially expanding the viral gene pool and challenging future control efforts. A One-Health surveillance framework that couples primate serology, high-resolution vector mapping, and real-time genomic sequencing can identify emergent transmission hot spots before they coalesce into sustained outbreaks [77].

Ultimately, safeguarding both human and wildlife health in West Africa will require moving beyond a strictly "urban dengue" mindset By quantifying sylvatic DENV-2 forces of infection in three non-human primate species and documenting that all three are frequently exposed to sylvatic DENV-2, our study provides the evidence base to craft vaccine strategies, land-use policies, and surveillance systems that are proportionate to the true, ecosystem-wide threat. Only through such integrated, forward-looking approaches can we hope to blunt the next wave of sylvatic DENV spillover.

## Supporting information

**S1 Fig. Seroprevalence to yellow fever virus (defined as PRNT$_{90}$ titer ≥1:20) by age and species in a subset of 298 non-human primates.**
(PDF)

**S2 Fig. Forces of infection (FOI) by species and year where panels show the forces of infection, $\lambda(a)$, for *C. sabaeus*, *P. papio*, and *E. patas*, respectively.** Points represent the proportion of seropositive animals per age and year, with associated confidence interval as vertical lines. Seroprevalence estimated from model using best fitting forces of infection (black line), with grey shaded bands representing bootstrap 95% confidence intervals. DENV PRNT$_{50}$ seropositivity was used to define prior infection.
(PDF)

**S1 Table Mixed effects logistic regression for DENV seropositivity.** Estimates from a mixed effects model with DENV PRNT$_{80}$ seropositivity as the outcome and monkey age, species, and year of collection as mixed effects with troop (same collection date and site) as the random effect.
(DOCX)

**S2 Table. Estimates of $R_0$ for each NHP species by year using varying age structure assumptions, and using PRNT$_{50}$ seropositivity to define prior infection.** The age distribution is assumed to be exponentially structured with rates equal to the mean lifespans reported in this study.
(DOCX)

## Acknowledgments

The authors thank Dr Mathilde Guerbois for invaluable help with the study. We gratefully acknowledge the impetus for this work provided by the NSF-RCN on Infectious Disease Evolution Across Scales (RCN-IDEAS) program.

## Author contributions

**Conceptualization:** Diawo Diallo, Mawlouth Diallo, Douglas M Watts, Amadou A Sall, Scott C Weaver, Kathryn A Hanley, Derek A.T. Cummings.

**Data curation:** Brenda Benefit, Kathryn A Hanley.

**Formal analysis:** Stephanie C Cinkovich, Benjamin M. Althouse, Matt DT Hitchings.

**Investigation:** Prudny Bonnaire-Fils, Ousmane M Diop, Ousmane Faye, El Hadji Abdourahmane Faye, Diawo Diallo, Bakary Djilocalisse Sadio, Abdourahmane Sow, Oumar Faye, Mawlouth Diallo.

**Methodology:** Benjamin M. Althouse, Derek A.T. Cummings.

**Writing – original draft:** Stephanie C Cinkovich, Matt DT Hitchings.

**Writing – review & editing:** Stephanie C Cinkovich, Benjamin M. Althouse, Matt DT Hitchings, Prudny Bonnaire-Fils, Ousmane M Diop, Ousmane Faye, El Hadji Abdourahmane Faye, Diawo Diallo, Bakary Djilocalisse Sadio, Abdourahmane Sow, Oumar Faye, Mawlouth Diallo, Brenda Benefit, Douglas M Watts, Amadou A Sall, Scott C Weaver, Kathryn A Hanley, Derek A.T. Cummings.

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
