## [Decision Letter · Decision Letter 0]

18 Sep 2025

Seroprevalence of dengue virus antibodies among multiple species of non-human primates in Senegal suggests that sylvatic dengue virus is maintained in non-primate reservoirs in this region

Dear Dr. Althouse,

Thank you for submitting your manuscript to PLOS Neglected Tropical Diseases. After careful consideration, we feel that it has merit but does not fully meet PLOS Neglected Tropical Diseases's publication criteria as it currently stands. Two of the three reviewers have pointed out aspects of the manuscript that could be improved, as indicated in their comments. Therefore, we invite you to submit a revised version of the manuscript that addresses the points raised during the review process.

Please submit your revised manuscript within 60 days Nov 17 2025 11:59PM. If you will need more time than this to complete your revisions, please reply to this message or contact the journal office at plosntds@plos.org. Please include the following items when submitting your revised manuscript:

We look forward to receiving your revised manuscript.

Kind regards,

Antonio Mas

Academic Editor

David Safronetz

Section Editor

Shaden Kamhawi

co-Editor-in-Chief

Paul Brindley

co-Editor-in-Chief

**Journal Requirements:**

At this stage, the following Authors/Authors require contributions: Stephanie C Cinkovich, Benjamin Althouse, Matt DT Hitchings, Prudny Bonnaire-Fils, Ousmane M Diop, Ousmane Faye, El Hadji Abdourahmane Faye, Diawo Diallo, Bakary Djilocalisse Sadio, Abdourahmane Sow, Oumar Faye, Mawlouth Diallo, Brenda Benefit, Douglas M Watts, Amadou A Sall, Scott C Weaver, Kathryn A Hanley, and Derek Cummings. Please ensure that the full contributions of each author are acknowledged in the "Add/Edit/Remove Authors" section of our submission form.

4) We notice that your supplementary Figures, and Tables are included in the manuscript file. Please remove them and upload them with the file type 'Supporting Information'. Please ensure that each Supporting Information file has a legend listed in the manuscript after the references list.

5) Please ensure that the funders and grant numbers match between the Financial Disclosure field and the Funding Information tab in your submission form. Note that the funders must be provided in the same order in both places as well.

State what role the funders took in the study. If the funders had no role in your study, please state: "The funders had no role in study design, data collection and analysis, decision to publish, or preparation of the manuscript.".

**Reviewers' Comments:**

Reviewer's Responses to Questions

**Key Review Criteria Required for Acceptance?**

**Methods:**

-Are the objectives of the study clearly articulated with a clear testable hypothesis stated?

-Is the study design appropriate to address the stated objectives?

-Is the population clearly described and appropriate for the hypothesis being tested?

-Is the sample size sufficient to ensure adequate power to address the hypothesis being tested?

-Were correct statistical analysis used to support conclusions?

-Are there concerns about ethical or regulatory requirements being met?

Reviewer #2: Methods: I appreciate the inclusion of both PRNT50 and PRNT80.

More details are needed in the Serology section (even if in the Althouse 2018 paper). For example, which DENV serotypes were used to investigate neutralization? All? If so, is there an availability of breaking down serology by serotype?

Reviewer #3: More detail on how PRNTs were performed is required

**Results**

-Does the analysis presented match the analysis plan?

-Are the results clearly and completely presented?

-Are the figures (Tables, Images) of sufficient quality for clarity?

Reviewer #2: It appears that the authors make the assumption that DENV2 is still the only DENV to circulate sylvatically in Senegal. The most recent paper cited for this is 2024, the data therein is from 2020 (with an additional non-exclusive report of a DENV2 sylvatic infection in 2021). This is a big assumption. If only DENV2 was tested, then the entire manuscript needs to be put into the context of DENV2 only.

Reviewer #3: Yes

**Conclusions**

-Are the conclusions supported by the data presented?

-Are the limitations of analysis clearly described?

-Do the authors discuss how these data can be helpful to advance our understanding of the topic under study?

-Is public health relevance addressed?

Reviewer #2: See comments in Results.

Reviewer #3: Several sections of the discussion are tangential to the data presented and could be summarized.

**Editorial and Data Presentation Modifications?**

Reviewer #2: (No Response)

Reviewer #3: NA

**Summary and General Comments**

Reviewer #2: This paper provides much needed data about the distribution of dengue in West Africa. This is a straightforward yet important study. The methods are sound the results are reported appropriately. However, there is some concern with over generalization.

Reviewer #3: The authors present a neat study discussing sylvatic DENV-2 seroprevalence and transmission dynamics in three monkey species in Senegal. These data are part of a larger study that characterized the sylvatic cycle of multiple arboviruses in Senegal. Results of CHIKV serologies were presented in a prior publication. However, I do not think that it is prudent to take a similar approach for DENV-2, particularly given the high degree of cross-reactivity seen among flaviviruses seen even with neutralization tests. My main suggestion to the authors is to consider reporting the YFV and ZIKV data here, or if there was little or no cross-reactivity seen in this cohort, then to explicitly state this as a reason for not factoring this into their models. My remaining major comments are outlined below.

- I am impressed by the scale of the field collection effort. Did the authors have any strategies to ensure no repeat sampling of individuals?

- The parent paper (Althouse 2018) does not show which virus strains (accession number) were used for PRNT. It would be important to do so at least for DENV-2, here. Also, PRNT50 and PRNT80 thresholds for seropositivity should be clearly defined.

- Given the concern for cross-protection/ cross-reactivity between flaviviruses, and knowing that YFV PRNTs were done on this cohort, the authors should strongly consider reporting these data here. If there was little or no cross-reactivity, then this can be easily shown. If there was cross-reactivity, could the models be adapted to consider this? What about ZIKV? Were PRNTs performed for ZIKV and if not, why not?

- In the discussion the authors note significant variability in trends among species, attributing this to troop size. Could they comment on usual troop sizes and behaviors of these different species?

- Is there any concern for spillback in these NHPs and could the DENV-2 seropositivity reflect this rather than true sylvatic DENV-2 infection? While I am not suggesting the authors perform PRNTs against different DENV-2 strains here, I think a brief mention of this potential limitation is warranted.

- The discussion is too long – for instance the paragraphs on amplification hosts (Lines 288-303), Aedes vectors (317-332), and changing land-use (Lines 359-368) present more like a review than a discussion of this study’s findings, and this ends up diluting the helpful discussion of other seroprevalence studies and the CHIKV findings in this cohort. Can the authors condense these paragraphs into a few salient points?

- Line 370-371: “By… highlighting how easily the virus can cross species and ecological boundaries,” I don’t think this is truly shown in this study?

- I cannot find the dataset using the provided doi, could the authors check that this is correct?

Minor comments:

Typo in Line 181: "estimated using by maximizing the likelihood" - remove "using"

Correct “DENV” with “DENV2” throughout as relevant (e.g., Lines 218, 219, 223, etc)

PLOS authors have the option to publish the peer review history of their article (what does this mean? ). If published, this will include your full peer review and any attached files.

**Do you want your identity to be public for this peer review?** For information about this choice, including consent withdrawal, please see our Privacy Policy .

Reviewer #2: No

Reviewer #3: No

**Figure resubmission:**
---

## [Decision Letter · Decision Letter 1]

19 Jan 2026

Dear Dr Althouse,

We are pleased to inform you that your manuscript 'Seroprevalence of dengue virus antibodies among multiple species of non-human primates in Senegal suggests that sylvatic dengue virus is maintained in non-primate reservoirs in this region' has been provisionally accepted for publication in PLOS Neglected Tropical Diseases.

Best regards,

Antonio Mas

Academic Editor

David Safronetz

Section Editor

Shaden Kamhawi

co-Editor-in-Chief

Paul Brindley

co-Editor-in-Chief

Reviewer's Responses to Questions

**Key Review Criteria Required for Acceptance?**

**Methods**

-Are the objectives of the study clearly articulated with a clear testable hypothesis stated?

-Is the study design appropriate to address the stated objectives?

-Is the population clearly described and appropriate for the hypothesis being tested?

-Is the sample size sufficient to ensure adequate power to address the hypothesis being tested?

-Were correct statistical analysis used to support conclusions?

-Are there concerns about ethical or regulatory requirements being met?

Reviewer #3: Yes

**Results**

-Does the analysis presented match the analysis plan?

-Are the results clearly and completely presented?

-Are the figures (Tables, Images) of sufficient quality for clarity?

Reviewer #3: Yes

**Conclusions**

-Are the conclusions supported by the data presented?

-Are the limitations of analysis clearly described?

-Do the authors discuss how these data can be helpful to advance our understanding of the topic under study?

-Is public health relevance addressed?

Reviewer #3: Yes

**Editorial and Data Presentation Modifications?**

Reviewer #3: None

**Summary and General Comments**

Reviewer #3: I am satisfied that the authors have addressed all my comments. The paper reads well and in my opinion is ready for publication.

PLOS authors have the option to publish the peer review history of their article (what does this mean? ). If published, this will include your full peer review and any attached files.

**Do you want your identity to be public for this peer review?** For information about this choice, including consent withdrawal, please see our Privacy Policy .

Reviewer #3: **Yes:** Christina Yek

---

## [Editor Report · Acceptance letter]

Dear Dr Althouse,

We are delighted to inform you that your manuscript, "Seroprevalence of dengue virus antibodies among multiple species of non-human primates in Senegal suggests that sylvatic dengue virus is maintained in non-primate reservoirs in this region," has been formally accepted for publication in PLOS Neglected Tropical Diseases.

Best regards,

Shaden Kamhawi

co-Editor-in-Chief

Paul Brindley

co-Editor-in-Chief
